# Cell-Based Therapy for Urethral Regeneration: A Narrative Review and Future Perspectives

**DOI:** 10.3390/biomedicines11092366

**Published:** 2023-08-24

**Authors:** Yangwang Jin, Weixin Zhao, Ming Yang, Wenzhuo Fang, Guo Gao, Ying Wang, Qiang Fu

**Affiliations:** 1Department of Urology, Shanghai Sixth People’s Hospital Affiliated to Shanghai Jiao Tong University School of Medicine, Shanghai Eastern Institute of Urologic Reconstruction, Shanghai Jiao Tong University, Shanghai 200233, China; jinyw_med@163.com (Y.J.);; 2Wake Forest Institute for Regenerative Medicine, Winston Salem, NC 27157, USA; 3Key Laboratory for Thin Film and Micro Fabrication of the Ministry of Education, School of Sensing Science and Engineering, School of Electronic Information and Electrical Engineering, Shanghai Jiao Tong University, Shanghai 200240, China

**Keywords:** cell therapy, stem cell, urethral regeneration, tissue engineering

## Abstract

Urethral stricture is a common urological disease that seriously affects quality of life. Urethroplasty with grafts is the primary treatment, but the autografts used in clinical practice have unavoidable disadvantages, which have contributed to the development of urethral tissue engineering. Using various types of seed cells in combination with biomaterials to construct a tissue-engineered urethra provides a new treatment method to repair long-segment urethral strictures. To date, various cell types have been explored and applied in the field of urethral regeneration. However, no optimal strategy for the source, selection, and application conditions of the cells is available. This review systematically summarizes the use of various cell types in urethral regeneration and their characteristics in recent years and discusses possible future directions of cell-based therapies.

## 1. Introduction

Urethral stricture is a pathological narrowing of the urethral lumen associated with excessive fibrosis of the epithelium and surrounding tissues [1]. As a common urological disease, urethral stricture can be caused by various factors such as trauma, inflammation, congenital malformation, and medically induced injury [2]. Urethral stricture can lead to urinary retention, bladder stones, fistula formation, urinary tract infection, hydronephrosis, and, in severe cases, renal failure, thus affecting quality of life, while treatment of urethral stricture also puts considerable pressure on the healthcare system [3,4]. Treatment of urethral strictures usually employs different repair strategies depending on the length, location, and cause of the injury [5]. Currently, urethroplasty is usually performed clinically using a graft in patients with long-segment urethral strictures/defects, recurrent strictures, or penile urethral strictures [6]. Grafts are usually autologous tissues, such as buccal mucosa, penile flap, and bladder mucosa, but they are limited in number and difficult to obtain. This treatment mode of sacrificing healthy tissues to repair lesions is also controversial because of the numerous complications associated with the donor site [7]. Therefore, reconstructive repair of the injured urethra remains challenging for urologists.

Tissue engineering is a derivative of regenerative medicine, which aims to create organs using cells, biomaterials, and engineering techniques [8]. Tissue-engineered grafts avoid the complications of autologous tissue collection and reduce patient pain. Over the past 30 years, using tissue engineering techniques to construct a tissue-engineered urethra has made great strides and is gradually being incorporated into urological practice. Single biomaterial scaffolds were first applied to urethral reconstruction, which provide good mechanical support and spatial structure for the migration of host cells and facilitate remodeling of the urethral tissue structure, achieving a certain degree of success in urethral repair. Some classical tissue engineering materials, such as small intestinal submucosa (SIS), bladder acellular matrix (BAM), and acellular dermal matrix, have been evaluated in several clinical trials. However, successful repair with a single scaffold is very dependent on a healthy urethral bed at the injury, adequate vascular distribution, and the absence of spongy fibrosis, which otherwise predisposes to chronic immune reactions, fibrosis formation and calcification, and graft shrinkage or restriction, for which a single scaffold is often inadequate to treat long defects [9]. A research shows that the maximum distance to repair the urethra using tubular acellular matrix grafts appears to be only 0.5 cm [10].

The natural healing process of the urethra involves interactions between multiple factors, such as intercellular contacts, secretions from resident and migrating cells, growth factors, cytokines, and various signaling pathways [11]. Therefore, many investigators have begun to explore cell-based regeneration strategies for long-segment urethral repair. A systematic evaluation showed that the long-term success rate of cell–scaffold material complex grafts inoculated with cells was 5.67 times higher than that of the scaffold material alone [12]. Transplanted cells promote rapid vascularization and re-epithelialization of the scaffold material at the graft repair site, reducing local inflammation and scar formation for better repair results [13,14,15]. Therefore, the involvement of cells is indispensable for urethral regeneration, particularly the repair of long-segment urethral defects.

To date, various cell types have been explored and applied in the field of urethral regeneration. However, no optimal strategy for the source, selection, and application conditions of the cells is available. This review systematically summarizes the use of various cell types in urethral regeneration and their characteristics in recent years and discusses possible future directions of cell-based therapies.

## 2. Non-Stem Cell-Based Regenerative Therapy

In most cases, the use of differentiated cells in urethral regeneration has been well established and includes mainly epithelial cells, smooth muscle cells, and endothelial cells.

### 2.1. Epithelial Cells

Epithelial cells are the key cells in urethra regeneration. A continuous layer of epithelial cells provides a barrier against corrosion effects and urinary fistula, thereby reducing inflammation and fibrous tissue deposition during the healing process [16].

Transitional epithelial cells of the bladder mucosa, mainly obtained by bladder biopsy, are considered the best candidates to reconstruct the epithelial cell layer of the urethra. Although structurally different from the compound columnar epithelium in the urethra, both the transitional epithelium in the bladder and the urethral epithelium are formed by p63+ cells from the urogenital sinus of the fetus [17], and both function as a barrier to urine in the urinary tract. In one study, the bladder mucosa was isolated and digested, and the obtained epithelial cells were cultured in supplemented CnT-07 for proliferation or CnT-02 with 1.07 mM/L CaCl_2_ for stratification [16]. The expanded autologous bladder epithelial cells were subsequently seeded on a type I collagen-based cell carrier, cultured in layers for 8 days, and labeled with PKH26. The collagen-based cell carrier grafts were then used to repair urethral strictures in minipigs. Immunofluorescence analysis confirmed the epithelial cell phenotype, junction formation, and differentiation at 2 weeks, and that the grafted cells were present at the repair site 6 months after surgery. No recurrence of the stricture was observed in the experimental animals at the final 6-month transplantation [16]. Another study using a rabbit model showed that scaffolds implanted with bladder epithelial cells supported epithelial integrity, stratification, and continuity with the normal urothelium [18]. Wang et al. [19] implanted bladder epithelial cells into human amniotic scaffolds to reduce rejection and improve the biocompatibility of the graft material. The results showed milder inflammatory cell infiltration, i.e., less accumulation of CD4 and CD8 cells, neutrophils, and other types of immune cells, in cell-seeded human amniotic scaffold grafts compared with the epithelial cell-free group. 

The number of cells obtained by biopsy is limited, and the procedure requires general anesthesia and is invasive. Bladder washing is a feasible alternative to harvesting viable autologous bladder epithelial cells in a non-invasive manner [20]. Amesty et al. [21] obtained autologous epithelial cells by bladder washing and seeded the submucosal matrix of acellular porcine small intestines to construct a tissue-engineered urethra. They found that the epithelial cell seeding group formed multilayered urothelial cells and successfully repaired urethral defects in rabbits. Epithelial cells collected by the bladder washing procedure have the same effect as biopsy acquisition, and it avoids donor site damage caused by biopsy, providing the possibility of obtaining cells several times for repeat procedures if needed [20].

Autologous urethral epithelial cells cannot, however, be obtained from patients with chronic inflammation of the urinary tract [22]. Oral mucosal cells exist in a humid physiological environment similar to the urinary tract and are resistant to a wet environment and infection because of the expression of beta-defensins and interleukin-8 in their membranes [23]. The differences between oral and urethral mucosae are minimal [24], and the collection of oral mucosal epithelial cells can be performed under local anesthesia in a simple and well-tolerated procedure. Therefore, autologous oral-derived epithelial cells are also an effective cell source for urethral regeneration. Huang et al. [25] used lingual keratinocytes seeded in a bacterial cellulose (BC) scaffold to treat rabbit urethral injuries. The cell–scaffold material composite group exhibited faster and more complete epithelial regeneration compared with the BC alone group. Lv et al. [26] cultured autologous lingual keratinocytes and seeded them on a novel three-dimensional (3D) scaffold composed of a combination of silk fibroin (SF) and BC and observed good regeneration of the urothelial cells in a dog urethral repair model. Oral mucosal epithelial cells have been developed to construct a tissue-engineered buccal mucosa for urethral repair and reconstruction, with encouraging results [2,27]. However, a potential limitation of oral mucosal epithelial cells is their low proliferative capacity and clonogenicity, hindering their large-scale expansion in vitro, which is required for clinical use.

Epidermal cells can also be isolated and expanded by minimally invasive methods from hairless skin, such as foreskin, and cultured to form a thick barrier that isolates urine. A study has demonstrated successful repair of rabbit urethral defects using a tubular acellular collagen matrix seeded with foreskin epithelial cells [28]. However, because of complications and malformations at the harvest site associated with a foreskin biopsy, recent studies have begun to explore the use of epidermal cells from other sites for urethral repair procedures. Rogovaya et al. [29] collected rabbit ear epithelial cells and cultured them to prepare cell sheets for rabbit urethral repair. The rabbits regained voluntary urinary function at 4–7 days postoperatively with no scarring or abnormal fistula formation in the urethra. Complete recovery of rabbit urothelial cells was observed at 45 days postoperatively in the grafted area, and the presence of a multilayered migrating epithelium was observed. In another study, Zhang et al. [30] obtained skin epidermal cells (SEC) from rabbit abdominal skin and constructed cryopreserved SEC-AM (amniotic membrane) urethral scaffolds for rabbit urethra repair, which also achieved good results. 

### 2.2. Mesothelial Cells

Epithelial cells have a low proliferative capacity and often require long culture cycles. Additionally, epithelial cells are unavailable under malignant conditions, a history of lichen sclerosis, or oral disease. Mesothelial cells have a higher proliferative capacity and plasticity than urothelial cells [31]. Studies have reported the successful use of mesothelial cell-lined grafts as urethral grafts, including peritoneal [32] and vaginal endografts [33]. Thus, mesothelial cells may be a suitable alternative to epithelial cells. Jiang et al. [34] seeded mesothelial cells onto autogenous granulation tissue to construct a mesothelium-lined compound graft for tubularized urethroplasty in male rabbits. Histologically, urothelial layers surrounded by increasingly organized smooth muscles were observed in seeded grafts. Conversely, myofibroblast accumulation and extensive scarring occurred in unseeded grafts. Although mesothelial cells of large omentum origin may have some advantages, long-term culture of mesothelial cells appears to be difficult because of early senescence. Thus, further studies on mesothelial cells are needed.

### 2.3. Smooth Muscle Cells

The well-developed smooth muscle layer enhances the mechanical properties of the urethra and maintains structural stability during stretching and urination, to some extent avoiding the occurrence of urethral strictures. Therefore, seeding smooth muscle cells (SMCs) for remuscularization of a tissue-engineered urethra is an effective method to possibly repair urethral injury while avoiding strictures. A bladder biopsy is the most common source of SMCs in urethral repair and reconstruction. In one study, bladder muscle tissue was clipped and digested, and a composite SMC scaffold was used to repair urethral defects in rabbits [35]. After 3 months, immunohistochemical examination showed a higher and well-arranged smooth muscle content in the SMC group compared with that in the control group, which resulted in a significantly lower rate of tubular obstruction and complications, including stone formation, urinary fistula, and urethral stricture incidence, in the SMC group. In a report by Lv et al. [36], earlier muscle regeneration was observed in the SMC-seeded scaffold group compared with the unseeded cell group. In another preclinical study, Niu et al. [37] seeded SMCs into a synthetic scaffold for rabbit urethral repair reconstruction and found that it promoted the regeneration of multilayered smooth muscle tissue, obtaining grafting results similar to those of autologous tissue. Similar success was subsequently achieved by their group in a dog model [38]. Ultimately, SMC composite scaffolds result in earlier, more mature muscle regeneration, thereby facilitating the avoidance of urethral strictures.

SMCs may also help support epithelial–mesenchymal interactions required for normal maturation of the urothelium [22,39]. In a preclinical study [14], autologous epithelial and smooth muscle cells were seeded in a tubular collagen scaffold and used for urethroplasty in 15 dogs. After up to 12 months of follow-up, computed tomography urethrograms showed a wide urethral caliber in animals treated with seeded cell grafts. Conversely, six control animals treated with unseeded scaffolds had blocked urethras. The seeded group showed superior epithelial tissue and muscle fiber formation, whereas the unseeded group showed fibrosis and few muscle fibers. In another study, autologous bladder epithelial and smooth muscle cells from nine male rabbits were expanded and seeded onto preconfigured tubular matrices constructed from acellular bladder matrices obtained from the lamina propria. Urethroplasties were performed with tubularized matrices seeded with cells in nine animals and matrices without cells in six animals. The urethrograms showed that animals implanted with cell-seeded matrices maintained a wide urethral caliber without strictures. Conversely, urethras with unseeded scaffolds collapsed and developed strictures [40]. Similarly, Lv et al. [26] successfully repaired urethral defects in dogs using a composite bilayer scaffold of lingual keratinocytes and lingual muscle cells. However, the low proliferative potential of smooth muscle cells and the relatively high trauma during primary cell collection make it difficult to obtain sufficient seeded smooth muscle cells.

### 2.4. Endothelial Cells

Blood vessels provide oxygen and nutrients to tissues and are necessary for tissue regeneration; therefore, vascularization of urethral grafts is an important step in the reconstruction of the urethra. Vascular endothelial cells have become a cell type of great interest. In one study, Heller and colleagues isolated human dermal microvascular endothelial cells from the foreskin to develop a pre-vascularized buccal mucosal substitute to repair urethral defects [41]. Their results showed successful pre-vascularization and the formation of dense capillary-like structures in the substitutes, which became functional vessels by anastomosis with host vessels after implantation into nude mice. Although endothelial cells have been shown to play a crucial role in promoting angiogenesis, harvesting primary endothelial cells remains challenging. Table 1 summarized the functions performed by differentiated cells in urethral regeneration.

## 3. Stem Cell-Based Regenerative Therapy

Stem cells are self-renewing and pluripotent, allowing them to differentiate into various cell types in the urothelial tissue in a specific microenvironment, and their paracrine secretion of various growth factors and bioactive cytokines stimulates the growth of nearby cells and has been shown to enhance angiogenesis and reduce fibrosis [42,43,44]. Therefore, stem cell therapy has great potential and has been a hot research topic in recent years [45,46]. Stem cells used for urethral regeneration mainly include bone marrow-derived stem cells (BMDSC), adipose-derived stem cells (ADSCs), and urine-derived stem cells (UDSC).

### 3.1. Pluripotent Stem Cells

Pluripotent stem cells include embryonic stem cells (ESCs) and induced pluripotent stem cells (iPSCs). ESCs are isolated from the inner cell mass of an embryonic blastocyst and have great clinical potential because they form cells of ectodermal, endodermal, and mesodermal origins. Blank et al. [47] established the first in vitro system to induce differentiation of mouse ESCs into SMCs by retinoic acid in 1995. Ottamasanthien et al. [48] described the differentiation of pluripotent stem cells towards the uroepithelium in a mouse model when ESCs were directed to the uroepithelial lineage through tissue reconstitution experiments with mouse embryonic bladder mesenchyme. However, ESCs have ethical restrictions because of embryo destruction for collection. iPSCs have similar regenerative and differentiation abilities to ESCs without the associated ethical controversies. Suzuki et al. [49] used a combination of PPAR (peroxisome proliferator-activated receptor)-γ agonists and EGFR (epidermal growth factor receptor) inhibitors, as well as FGF10 and transwell cultures, to demonstrate directed differentiation of iPSCs into mature stratified bladder epithelium. However, iPSCs have some concerns, including low reprogramming and differentiation efficiencies and potential tumorigenicity [22].

### 3.2. Bone Marrow-Derived Stem Cells

Mesenchymal stem cells differentiate into various cell and tissue types. As a type of mesenchymal stem cell, BMDSCs can differentiate into urothelial cells and bladder SMCs in vitro and in vivo, and initial success in urethral regeneration has been achieved. Demirel et al. [50] evaluated the effects of BMDSC injection in a rat model of posterior urethral injury and demonstrated that BMDSC treatment significantly reduced the development of fibrosis in a uroepithelial injury model. In addition to BMDSC injection alone, a composite scaffold may promote their further differentiation and improve pro-regenerative and tissue fusion effects. Zhang et al. [51] compared bladder regeneration of BMDSC-seeded and bladder SMC-seeded SIS scaffolds. Histological evaluation showed that SIS grafts implanted with BMDSCs showed solid smooth muscle bundle formation throughout the graft at 10 weeks after surgery, which was similar to the results achieved in the bladder SMC cell-seeded SIS group. Another study compared the therapeutic effects of a BMDSC composite scaffold and an autologous oral mucosa graft for urethral reconstruction [52]. To track BMDSCs in vivo, they were labeled with superparamagnetic iron oxide nanoparticles. Twelve weeks of follow-up revealed that the BMDSC composite scaffold formed good fusion with the surrounding urethral tissue, and histology showed less fibrosis and inflammatory cell infiltration of lymphocytes, histiocytes, and plasma cells in the experimental group compared with those in the autologous oral mucosa graft group. Interestingly, nanoparticle-labeled BMDSCs were detected in the urethral epithelium and muscle layer, which colocalized with uroepithelial cytokeratin markers AE1 and AE3, suggesting differentiation of inter-BMDSCs into new urethral epithelium.

Interactions of BMDSCs with other cells may further promote tissue regeneration. A study evaluated the effect of a cell/scaffold composite graft consisting of human BMDSCs with CD34+ hematopoietic stem/progenitor cells on modulating inflammation and wound healing in a rodent model of substitution urethroplasty [53]. The urethra of cell-seeded animals showed 1.3- and 1.7-fold reductions in levels of the inflammatory cytokine TNF (tumor necrosis factor)-α and neutrophil migration, respectively, within 2 days after surgery compared with unseeded animals. This early difference in the inflammatory response between seeded and unseeded animals became more pronounced over time, eventually leading to 4.6- and 8.8-fold reductions in the levels of TNF-α and neutrophil migration, respectively, at 4 weeks. Histologically, changes in vascular profiles were evident, with initially small and numerous vessels developing into larger, more mature vessels during the healing process in the seeded group. Conversely, the control group showed no such progression. On the basis of AM, Chen et al. [54] seeded BMDSCs and endothelial progenitor cells (EPCs) to successfully repair long-segment circumferential urethral defects in a canine model. The presence of BMDSCs promoted EC survival, proliferation, and migration and contributed to EPC recruitment for angiogenesis. 

Despite these positive results, BMDSC acquisition requires bone marrow aspiration, and this highly invasive method of acquisition limits their use.

### 3.3. Adipose-Derived Stem Cells

ADSCs that are widely distributed and abundant in the human body are readily available because >400,000 liposuction surgeries are performed annually for cosmetic or medical purposes, with up to 3 L of liposuction fluid discarded after each procedure [55]. Additionally, their proliferation efficiency and potential for multidirectional differentiation have been extensively studied. In an epithelial-specific microenvironment, ADSCs display a stratified epithelial-like morphology with increased expression of epithelial-specific proteins and eventually differentiate into uroepithelial-like cells [56,57,58]. Urethral reconstruction using post-epithelial induction ADSC composite BAM showed that post-induction ADSC composite BAM replantation formed epithelial-like structures at the replacement site in vivo and reduced scar contracture and stricture formation in the reconstructed urethral segment to some extent [59]. ADSCs can also differentiate towards smooth muscle in response to mechanical or specific microenvironmental stimuli [60,61]. Fu et al. [60] used mechanical stimulation to differentiate ADSCs towards smooth muscle cells and subsequently seeded the cells in a PGA (polyglycolic acid) scaffold that was applied with good results in a dog urethral repair model. The differentiated ADSCs constituted an engineered urethra that adapted to the mechanical extension generated by the urinary stream and helped to reduce the incidence of urethral strictures. 

Prevention of fibrosis and reduction of scarring play a crucial role in urethral repair, and the anti-fibrotic effects of ADSCs have been widely explored in recent years. One study evaluated the anti-urethral fibrosis effect of ADSCs [62]. The urethral walls of rats were incised and injected with the fibrosis inducer transforming growth factor-β1. One day later, ADSCs were injected into the urethral walls of rats in the ADSCs group. After 4 weeks, the rats were evaluated histologically and functionally. Compared with the control group, the ADSCs-treated group showed a significant increase in single-void volume, urine flow rate, bladder compliance, and bladder volume with prolonged voiding intervals. Moreover, the overall structure of the spongious urethra, and the collagen and elastin contents of the penile shafts approximated the normal urethra. Because of the complex histopathological microenvironmental changes at the injury site, cell transplantation alone may not be sufficient. MiR-21 has been proven to assist stem cell differentiation and paracrine secretion. Recent studies have shown that miR-21 also plays a major role in skin fibrosis [63,64]. Feng et al. [65] explored whether miR-21 modification improved the efficacy of ADSCs against urethral fibrosis and thus limited the recurrence of urethral strictures. They established miR-21-modified ADSCs by lentivirus-mediated transfer of pre-miR-21 and GFP reporter genes. In vitro results showed that miR-21 modification increased angiogenic gene expression in ADSCs and enhanced their antioxidant effects against reactive oxygen species damage. In vivo results demonstrated that the miR-21 modification contributed to improved urodynamic parameters and better formation of epithelial and muscle layers compared with the group injected with ADSCs alone. This study validated the potential of miR-21 to improve the anti-urethral fibrosis of ADSCs, but further studies are needed to determine long-term efficacy. A recent study showed that human mesenchymal stem cells inhibit fibroblast activation and the associated inflammatory responses via miR-146a in exosomes, which may also contribute to the mechanism of ADSC-mediated inhibition of urethral fibrotic strictures [66]. Notably, however, because most human urethral fibrosis is in advanced and chronic stages, further evaluation of the effect of stem cell injection therapy on established and recurrent urethral fibroses is important to translate this therapy into the clinic.

ADSC composite scaffolds have also undergone significant improvements. In one study, autologous ADSCs from dogs were grown and seeded onto a premade acellular arterial matrix [67]. Seeded scaffolds were used to repair surgically produced urethral defects in six male dogs, and the results were compared with those of six control animals treated with the acellular arterial matrix. Serial urethrography was performed postoperatively at 1 and 3 months. All six animals in the experimental group had a wide urethral caliber without any signs of stricture. Conversely, three animals in the control group showed urethral strictures. Similar success was achieved in a rabbit urethroplasty model using an ADSC-seeded SF scaffold [68]. Compared with the application of SF alone, the composite group showed a milder inflammatory response and more vascular and smooth muscle tissue formation. Yang et al. [69] constructed a composite hydrogel patch accommodating ADSCs for rabbit urethral repair by multilayer 3D bioprinting. Compared with the unseeded ADSC control group, the seeded ADSC group showed reductions in bulk scarring and the urographic obstruction rate, and pathology showed that the introduction of ADSCs significantly reduced the collagen fiber content. Notably, evidence that the therapeutic advantage of ADSCs lies not only in their multipotency but also in their trophic and paracrine functions is growing [70]. Therefore, exploring methods to increase the paracrine activity of ADSCs may further promote tissue regeneration. Under hypoxic preconditioning, ADCSs have enhanced paracrine activity, proliferation, and survival [71,72]. Modulation of ADSC paracrine factors is also an effective method to enhance the pro-regenerative effect of ADSCs. As a protein that broadly affects the FGF signaling pathway, FGFR2 (Fibroblast growth factor receptors 2) is closely associated with the development and repair of the urinary tract. Therefore, targeted modification of ADSCs to overexpress FGFR2 may contribute to their secretory function and reparative effects. Zhu et al. [73] constructed a composite scaffold of ADSCs overexpressing FGFR2 (Figure 1a). The ADSC modification promoted the secretion of angiogenic factors and enhanced their proliferation and migration abilities, which promoted tissue angiogenesis and regeneration, resulting in excellent reparative effects in a rabbit urethral injury model. TIMP-1, which is highly expressed in urethral scar tissue, plays a crucial role in the urethra stricture [74,75,76]. Sa et al. [77] performed the first miRNA modification of epithelial differentiated adipose-derived stem cells (E-ADSCs) to reduce expression of the profibrotic factor TIMP-1 and evaluated their effectiveness for urethral repair. The modified E-ADSCs seeded in BAM inhibited fibrosis in urethral tissue, leading to a wider urethral caliber. In another study, hypoxia-treated ADSCs were seeded in porous nanofiber scaffolds and used to repair rabbit urethral defects (Figure 1b–d). The in vivo results showed that hypoxia-preconditioned ADSCs combined with scaffolds led to a larger urethral lumen diameter, preserved urethral morphology, and enhanced angiogenesis compared with normoxia-preconditioned ADSCs [78].

The extracellular matrix (ECM) of ADSCs retains its secreted biological factors, which facilitate tissue regeneration, and is a good alternative source of decellularized matrix [79]. In urethral repair and reconstruction, decellularized tissue matrices such as SIS and BAM have been used with some degree of success in preclinical and clinical studies. However, the limited amount of autologous tissue-derived ECM is prone to surgical complications at the donor site, and there is a risk of disease transmission and ethical issues [80,81]. Zhou et al. [82] fabricated an ADSC-ECM using a repeated freeze–thaw cycle, Triton X-100, and SDS decellularization. Oral mucosal epithelial cells had a higher survival rate on ADSC-ECM compared with SF and formed a continuous layer of epidermal cells. Compared with SIS, mononuclear macrophage infiltration was less in ADSC-ECM when implanted subcutaneously into rats. Additionally, mRNA expression of cytokines, such as IL-4 and IL-10, was significantly higher in ADSC-ECM than in SIS at 3 weeks post-implantation.

### 3.4. Urine-Derived Stem Cells

UDSCs are a subpopulation of stem cells isolated from human urine that differentiate into various cell types, including SMCs, epithelial cells, and endothelial cells [83,84,85]. UDSCs express smooth muscle-specific proteins, including α-smooth muscle action, desmin, and myosin when cultured under PDGF (platelet-derived growth factor)-BB and TGF (transforming growth factor)-β1 induction, and uroepithelial-specific proteins AE1, AE3, and E-cadherin when cultured under epidermal growth factor induction (Figure 1e,f) [86]. They share many biological properties with mesenchymal stem cells, such as potent paracrine effects and immunomodulatory capacity. However, compared with other stem cells, autologous UDSCs can be harvested by a simple, safe, low-cost, and non-invasive procedure [84,87]. Additionally, up to 75% of fresh UDSCs can be safely persevered in urine for 24 h and retain their original stem cell properties [88]. Interest in UDSCs has increased over the past decade because of their great potential for regenerative medicine applications.

In one study [89], human UDSCs were extracted, and five induction methods were used to optimize their differentiation towards the uroepithelium. Induced cells were assessed for expression of gene and protein markers of urothelial cells by RT-PCR, Western blotting, and immunofluorescence staining. The barrier function and ultrastructure of tight junctions were assessed by permeability assays and transmission electron microscopy. Phenotypic and functional characteristics similar to those of native urothelial cells were observed in induced UDSCs. Additionally, multilayered urothelial tissue had formed 2 weeks after the inoculation of induced UDSCs on the intestinal submucosal matrix. Liu et al. [87] obtained rabbit autologous UDSCs from urine and bladder washings and seeded them on SIS to repair urethral defects in a rabbit model. It was found that autologous UDSCs differentiated into urothelial cells and SMCs in the natural urethral environment and performed corresponding functions. Compared with the unseeded cell group, the urethral caliber, urethral regeneration rate, smooth muscle content, and vascular density were significantly improved in the autologous UDSC-seeded SIS group.

**Figure 1 biomedicines-11-02366-f001:**
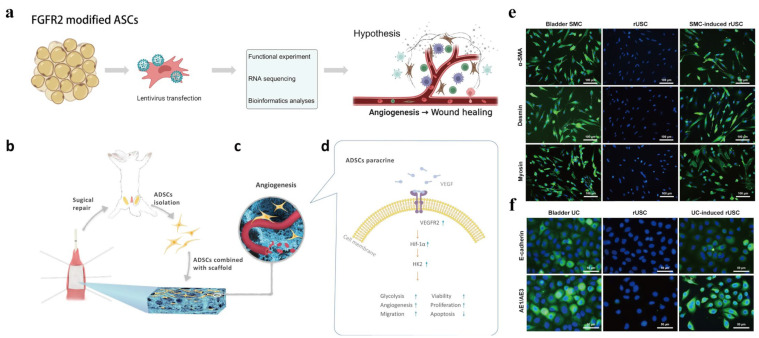
Methods to promote paracrine secretion in ADSCs and differentiation of UDSCs. (**a**) ADSCs overexpressing FGFR2. Reprinted (adapted) with permission from Ref. [73]. Copyright ©2022, Springer Nature. (**b**) ADSCs isolated from bilateral inguinal adipose tissues and then seeded on a scaffold to repair a urethral defect. (**c**) Hypoxia-preconditioned ADSCs grow in macropores and promote angiogenesis. (**d**) Mechanism of hypoxia-preconditioned ADSCs secreting more VEGFA to promote expression of VEGFR2, HIF-1α, and HK2 to upregulate glycolysis. Reprinted (adapted) with permission from Ref. [78]. Copyright ©2020, Springer Nature. (**e**) UDSCs expressing smooth muscle-specific markers desmin, myosin, and α-SMA assessed by immunofluorescence after culture in myogenic differentiation medium containing TGF-β (2.5 ng/mL) and PDGF (5 ng/mL) for 14 days. (**f**) UDSCs expressing urothelial-specific markers AE1 and AE3 assessed by immunofluorescence were significantly increased at 14 days after culture in uroepithelial differentiation medium with epidermal growth factor (30 ng/mL) compared with non-induced rUSCs. Reprinted (adapted) with permission from Ref. [86]. Copyright ©2018, Springer Nature.

UDSCs are innate to the urinary tract and thus have better histocompatibility and can survive in urine similarly to healthy urothelial cells. Therefore, UDSCs have great application potential and are a good source of tissue-engineered urothelial seed cells. However, the current literature concerning UDSC applications is limited, and further studies are needed.

### 3.5. Other Stem Cell Types

EPCs participate in vascular remodeling and angiogenesis by migrating to sites where blood vessels are needed [90]. Chen et al. [54] isolated EPCs from bone marrow, seeded them into AM, and successfully repaired a 3 cm long segmental circumferential urethral defect in a canine model. Complete vasculature development was observed in animals that received scaffolds seeded with EPCs, in contrast to those that received unseeded scaffolds or a sham operation [54]. In another preclinical study, more pronounced angiogenesis was observed in the EPC-seeded group compared with the control group [91].

Human amniotic fluid stem cells (HAFSCs) are multipotent stem cells of mesenchymal origin extracted from amniotic fluid [92]. HAFSCs can differentiate into various tissue types, such as skin, cartilage, and kidneys. Because stem cell extraction does not require the destruction of human embryos, the use of HAFSCs is less controversial. Kang et al. [93] demonstrated that microenvironmental changes induced by bladder-specific culture medium were sufficient to induce differentiation of HAFSCs into urothelial cells. Therefore, HAFSCs may be an effective alternative source of urothelial cells. However, further in vivo studies are needed to verify their effectiveness in urethral regeneration.

Recently, human amniotic mesenchymal stem cells (hAMSCs) derived from the amniotic membrane have attracted attention [94]. Similar to BMSCs and ADSCs, hAMSCs are multipotent, highly proliferative, and immunotolerant [95,96]. A major advantage of hAMSCs is that they are readily available, which eliminates the invasive procedures and ethical issues of cell harvesting [94]. Lv et al. [97] used hAMSC composite scaffolds to repair urethral defects in rabbits. The results showed a significantly lower incidence of urethral stricture, urinary fistula, and complications compared with the unseeded cell group. The seeded cell group formed a multilayered mucosa similar to normal urethral tissue after 12 weeks. Table 2 summarized the functions performed by stem cells in urethral regeneration.

## 4. Cell Sheet Technology

The cell sheet technique may be a promising novel approach in the field of urethral regeneration. A cell sheet retains the extracellular matrix with active factors, which facilitates local cell proliferation and regeneration [98]. It does not require enzymatic digestion and has a higher cell viability rate. A study compared the effectiveness of the cell sheet technique with the cellular perfusion technique for recellularization of the urethral decellularized stroma and found that the cell sheet technique achieved more effective recellularization as assessed by histopathology [99]. Alternatively, a cell sheet retains tight junctions between cells, and its dense structure restores the smooth and watertight properties of the urethral mucosa, ensuring unobstructed urination and preventing urine leakage, mimicking the native urethral epithelium. Liang et al. [100] successfully repaired rabbit urethral mucosal defects using autologous ADSC sheets (Figure 2a). The cell sheets were labeled with indocyanine green, and second near-infrared fluorescence imaging was performed to track ADSC sheets in vivo. Histological analysis showed that, in the ADSC sheet group, continuous epithelial cells covered the urethra at the graft site, and a large number of vascular endothelial cells were seen. In the cell sheet-free group, there was no continuous epithelial cell coverage at the urethral repair site, and expression of the proinflammatory factor TNF-α was increased. Similar success was achieved with transplanted skin epithelial cell membranes in a rabbit urethral injury model [29]. Cell sheets are more manipulatable and can be arranged and compounded in accordance with the tissue anatomy and cellular composition to form a biomimetic material scaffold with a 3D structure. Mikami et al. [101] collected oral tissues by biopsy, isolated mucosal and muscle cells, and cultured epithelial and muscle cell sheets, respectively. After 2 weeks, the two cell sheets were ligated and tubularized to construct two layers of tissue-engineered urethra and transplanted into urethral defects in dogs (Figure 2b). Histological analysis at 12 weeks after grafting demonstrated that urethras in the cell sheets group had formed stratified epithelia, and a well-vascularized submucosa was observed under the epithelial layers with cells. The urethrogram at 12 weeks revealed maintenance of a wide urethral caliber without stricture, leakage, or dilatation in the cell sheet group. The urethral histological structure consists of mucosa, submucosa, and muscles from the inside to the outside. Guided by the histological features of the urethra, Zhou et al. [102] chose various seed cells (oral mucosal epithelial cells, oral mucosal fibroblasts, and ADSCs) to construct the corresponding cell sheets and labeled the cells with ultrasmall superparamagnetic iron oxide at optimized concentrations (Figure 2b). Biomimetic urethral subcutaneous grafts significantly increased the urethral vascular density after 3 weeks and were subsequently used to repair urethral defects in dogs. After 3 months of urethral replacement, the biomimetic urethra maintained a three-layer structure and functions in a manner similar to the natural urethra. In another study, ADSC sheet self-assembled scaffolds supported the adhesion and growth of urothelial cells and SMCs. Seeding with both cell types is expected to lead to the development of a fully functional human urethra [103]. Notably, however, cell sheets tend to have long culture cycles and poor mechanical properties. To overcome these limitations, Zhang et al. [30] explored the possibility of using cryopreserved epithelial cell sheets combined with AM for rabbit urethral repair. The addition of AM enhanced the mechanical properties of epithelial cell sheets and reduced cell damage caused by cryopreservation. Histological examinations after 1 month showed that the urethral epithelium had completely regenerated with slight collagen deposition and adequate vascular regeneration under the mucosal layer.

The advantage of cell sheet technology is that it removes the influence of scaffold material degradation [98]. Notably, however, the current cost required to produce patient-derived cell sheets limits their widespread use.

**Figure 2 biomedicines-11-02366-f002:**
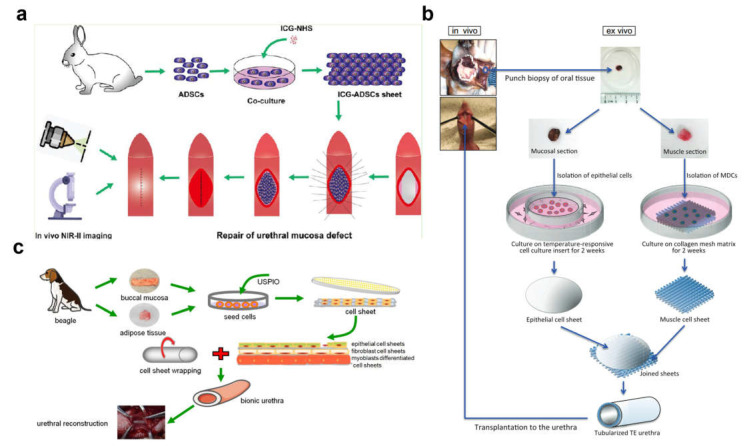
Cell sheet technology. (**a**) Repairing rabbit urethral mucosal defects using autologous ADSC sheets. ADSC sheets were labeled with indocyanine green (ICG), and second near-infrared (NIR-II) fluorescence imaging was performed to track ADSC sheets. Reprinted (adapted) with permission from Ref. [100]. Copyright © 2022, American Chemical Society. (**b**) Epithelial and muscle cell sheets were ligated and tubularized to construct two layers of tissue-engineered urethra and transplanted into urethral defects in dogs. Reprinted (adapted) with permission from Ref. [101]. Copyright © 2023 American Urological Association Education and Research, Inc. (**c**) Three types of cell sheets (oral mucosal epithelial cells, oral mucosal fibroblasts, and ADSCs) were cultured to construct a biomimetic urethra. Reproduced under terms of the CC-BY license [102]. Copyright © 2023 Ivyspring International Publisher.

## 5. Clinical Studies

Urethroplasty is the standard treatment for long-segment urethral strictures [104], which has a high success rate and long-term durability, but it is highly invasive, technically demanding, and has a steep learning curve [105,106]. Patients undergo endoscopic treatment several times before urethroplasty. Therefore, researchers have tried to improve endoscopic treatment to further improve the results of urethrotomy or dilation. Although treatments by injection of various drugs have been proposed, no benefit has been shown in clinical trials [107]. However, cell-based therapies offer promising directions that have shown initial success. Vaddi et al. [108] reported BEES-HAUS (buccal epithelium expanded and encapsulated in a scaffold-hybrid approach to urethral stricture). Autologous cultured buccal epithelial cells that are expanded and encapsulated in TGP scaffolds are implanted at the stricture site after a wide endoscopic urethrotomy to form an epithelial layer. The procedure was successful in four of six patients, yielding more than a 3-year recurrence-free interval. The subsequent use of this method in a rabbit model confirmed the success of buccal mucosal epithelial cell transplantation [109,110]. Recently, Scott et al. [107] proposed a novel method of urethral stricture treatment using liquid buccal mucosal grafts to augment direct vision internal urethrotomy. The results showed a 67% transplantation rate in the treatment group, but its treatment success rate was not statistically significant compared with the control group. In another prospective human study, encouraging results were obtained by buccal mucosal epithelial cell transplantation. Kulkarni et al. [111] evaluated the safety and efficacy of AALBECs (autologous adult living cultured buccal epithelial cells) in the treatment of male bulbar urethral strictures. Approximately 1 × 1.5 cm of oral mucosal tissue was collected from the inner cheek under local anesthesia, from which the epithelial layer was isolated and cultured, expanded in vitro, tested, and prepared as a suspension of 2.5 million cells/0.4 mL DMEM (Dulbecco’s modified Eagle’s medium) per vial, which was then injected into the stricture site after cystoscopic dissection of the stricture. After AALBEC treatment, patients showed a decrease in voiding time and urinary flow time (*p* < 0.05) and a 90.5% reduction in the mean AUA (American Urological Association) symptom index, and no patient required surgery within 24 weeks after treatment. These results demonstrate that buccal epithelial cell transplantation may be an effective alternative to urethrotomy and dilatation and may be a novel treatment option for urethral reconstruction. However, the results need to be further substantiated in large, well-designed studies.

Buccal mucosa is one of the most widely used grafts to repair urethral strictures. However, it is limited in number and prone to donor site complications [112,113]. The development of tissue-engineered buccal mucosa (TEBM) may overcome the limitations of autologous oral mucosa grafts. In 2008, Bhargava et al. [114] first reported the results of TEBM in a clinical trial. Keratinocytes and fibroblasts were isolated and cultured, seeded on a sterilized donor’s de-epidermized dermis, and maintained at the air–liquid interface for 7–10 days to obtain TEBM grafts. Five patients with urethral strictures secondary to lichen sclerosus underwent TEBM urethroplasty. At a mean follow-up of 33.6 months, three of the five patients had a patent urethra, while the other two developed fibrosis and constriction. This study demonstrated the potential of TEBM for the treatment of urethral strictures. MukoCell is a commercial tissue-engineered graft containing autologous oral epithelial cells on a collagen matrix [115]. Ram-Liebig et al. [27] reported the results of a multicenter, prospective, observational trial using an industrial tissue-engineered oral mucosa graft with market authorization in Germany (MukoCell) in 99 men. Using conservative Kaplan–Meier assessment, no stricture recurrence was observed in 67.3% (95% CI 57.6–77.0) of men at 12 months after the operation or in 58.2% (95% CI 47.7–68.7) of men at 24 months. These results were broadly similar to buccal mucosal urethroplasty [113]. In another retrospective multicenter study, 38 patients with recurrent strictures (median stricture length of 5 cm) underwent MukoCell urethroplasty with a median follow-up of 55 months, resulting in 32 (84.2%) successful treatments. No local or systemic adverse effects due to the engineered material were observed [116]. TEBM offers a safe and effective treatment opportunity for patients with urethral strictures, but it needs to be validated in long-term clinical trials with large samples. At present, regulatory, legal, and financial issues are major factors that restrict and impede the widespread use of these technologies in many countries [117].

## 6. Future Perspectives

In the past decades, urethral tissue engineering has made great strides, and the development of biomaterials has been gradually integrated into urological practice. However, the use of biomaterials alone has many shortcomings in practical use, especially for long segments of urethral strictures. Using various types of seed cells in combination with biomaterials to construct a tissue-engineered urethra provides a new treatment method to repair long-segment urethral strictures.

Differentiated cells were first explored for application in the field of urethral regeneration, and many researchers have attempted to apply them in clinical studies. One of the most popular is epithelial cells because the continuous epithelial layer plays an important role in resisting urine, effectively avoiding wound erosion and urethral fistula. Clinical studies have shown that buccal mucosal epithelial cell injection alone effectively prevents urethral fibrosis, and TEBM constructed with buccal mucosal epithelial cells has also produced good results. In addition to epithelial cells, SMCs and endothelial cells also play an important role in urethral wound healing, but these two cell types are more difficult to be obtained and have low cell proliferation potential. Thus, they are unsuitable for clinical research and practice until these issues are resolved.

Stem cells have been a hot research topic in recent years and, in many respects, have advantages over differentiated cells. Stem cells expand in vitro and are highly plastic, differentiating into specific cell types in urethral tissue in a specific microenvironment. In addition to secreting paracrine growth factors to enhance angiogenesis and reduce fibrosis, stem cells promote tissue regeneration by secreting active factors to recruit endogenous cells [118,119]. More importantly, they possess immune escape properties while allowing the use of allogeneic sources [120], which eliminates the need for autologous cell harvesting and expansion, thereby reducing the overall cost and duration of treatment [121]. Among them, pluripotent stem cells have received great attention in the field of regenerative medicine, but moral and ethical issues have limited their application. Most current studies have favored MSCs, especially ADSCs, which are a good cell source for urethral regeneration because of their abundance, easy access, and high proliferation efficiency, and promising results have been obtained in many studies. UDSCs of urinary tract origin are also very attractive. Unlike other stem cells, UDSCs can be obtained non-invasively (i.e., from urine). One study [122] has compared the stem cell properties and differentiated abilities of UDSCs and ADSCs collected from the same patient. Population-doubling time colony formation assays showed that UDSCs possessed a greater growth capacity. Additionally, analysis of multipotent differentiation (myogenic, neurogenic, and endothelial cells) showed that UDSCs were better than ADSCs. However, the current conditions for stem cell differentiation are stringent and need to be further optimized to achieve more stable and mature differentiation. Moreover, because stem cells are affected by the microenvironment, the effects of microenvironmental stimuli, such as mechanical forces, pH, signaling molecules, and oxygen levels, in the urethra on stem cells should be fully studied and considered before proceeding to human trials.

In tissue engineering, the biological microenvironment affects cell survival, colonization, and differentiation. Therefore, cell culture conditions, delivery methods, and biomaterial types and structures are also very important factors that need to be further explored. A reasonable combination of these factors may facilitate the full utilization of cells to construct a structurally and functionally complete biomimetic urethra more precisely and further improve the urethral reparative effect. An important direction in the field of urethral regeneration in the future may be cell sheet technology. Cell sheet technology achieves more effective recellularization of biological materials or damaged tissues. It also preserves the tight connections between cells and is more conducive to constructing a biomimetic urethra. Notably, however, the fabrication time and cost of cell sheets are obstacles that need to be overcome.

## 7. Conclusions

Cell-based therapies are promising in the field of urethral regeneration. To date, several cell types have been explored and applied in the field of urethral regeneration, but there is no optimal strategy for the source, selection, and application conditions of the cells. In this review, we summarized the various cell types applied to urethral regeneration and discussed their characteristics and conditions of application. We suggest that stem cells are a promising option for the future and that ADSCs and urinary tract-derived UDSCs may be the best cell sources for cell-based therapies. However, the differentiation conditions of stem cells need to be further optimized. The key to successful urethral regeneration lies in the construction of well-organized and functional biomimetic urethral grafts, which cannot be achieved without a rational combination of cell and tissue engineering technologies, including scaffolds and cell delivery techniques.

## Figures and Tables

**Table 1 biomedicines-11-02366-t001:** Summary of the functions performed by differentiated cells in urethral regeneration.

Cell Type	Source	Function in Urethral Regeneration	References
Mucosal epithelial cells	Bladder mucosa	Support epithelial integrity, stratification, and continuity with normal urothelium; reduce potential rejection reactions; and improve the biocompatibility of the graft material.	[16,18,19,21]
	Oral mucosa	Promotion of urethral epithelial regeneration; participation in the construction of tissue-engineered buccal mucosa (TEBM).	[2,25,26,27]
	Skin/foreskin	Form a thick barrier to isolate urine.	[19,30]
Mesothelial cells	Peritoneal/vaginal endothelial	Act as a substitute for epithelial cells.	[34]
Smooth muscle cells	Bladder	Promote earlier, more mature regeneration of urethral smooth muscle; enhance the mechanical properties of grafts; and support the epithelial–mesenchymal interactions required for normal maturation of the urothelium.	[14,22,26,35,36,37,38,39,40]
Endothelial cells	Foreskin	Promote tissue angiogenesis and graft vascularization.	[41]

**Table 2 biomedicines-11-02366-t002:** Summary of the functions performed by stem cells in urethral regeneration.

Cell Type	Source	Function in Urethral Regeneration	References
Pluripotent stem cells	Human embryos	Differentiate to any cell type in the urethra.	[47,48,49]
	Reprogrammed cells from adult tissues		
BMDSCs	Bone marrow	Differentiate into urothelial cells and bladder SMCs; reduce fibrosis and inflammation; and interact with other cells to further promote tissue regeneration.	[50,51,52,53,54]
ADSCs	Adipose tissue	Differentiate into urothelial cells and SMCs; prevent fibrosis and reduce scarring; promote regeneration of vascular and smooth muscle tissues; and reduce the inflammatory response. Paracrine function promotes regeneration.	[56,57,58,59,60,61,62,63,64,65,66,67,68,69,70,71,72,73,77,78]
UDSCs	Urine	Differentiate into urothelial cells, SMCs, and endothelial cells; promote regeneration of vascular and smooth muscle tissues; secrete various growth factors; and promote vascularization.	[83,84,85,86,87,88,89]
EPCs	Venous blood and bone marrow	Involved in vascular remodeling and angiogenesis.	[54,90,91]
hAFSCs	Amniotic fluid	Differentiate into urothelial cells.	[92,93]
AMSCs	Amniotic membrane	Promotes regeneration of the urethral epithelium.	[97]

## Data Availability

As no new data were created in this article, data sharing is not applicable to this article.

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
