# Peer review of "Cell-Based Therapy for Urethral Regeneration: A Narrative Review and Future Perspectives"

_biomedicines, 2023, doi:10.3390/biomedicines11092366_

Round 1

Reviewer 1 Report

well written comprehensive manuscript on usage of tissue engineering in the field of urethral stricture surgery 

Title - the title is misleading - this is not a systematic review, but a narrative one - major

Introduction - raw 35-38 - this definitive statement is highly controversial. In this reviewer`s opinion it should be true only in the most complicated forms of urethral disease with more than 10-12 cm urethral length needed to be replaced.  

paragraph 2 - Non-stem cells - nicely written paragraph presenting in depth the advantages and weaknesses of differentiated cells

paragraph 3  - stem cells 

raw 433 - mesenchymal - maybe a typo for membrane

paragraph 4 - cell sheet - nicely written paragraph on this novel technology

paragraph 5 - clinical studies - in this reviewer`s opinion one of the most important parts of the study - comprehensive analysis of present level of clinical science on the subject

Paragraph 6 and 7 - nicely written analysis on future directives in development and conclusions

Regarding all the aforementioned, my recommendation is to accept this manuscript for publication after satisfactory comments and appropriate modification by the authors on the abovementioned issues.

Reviewer 2 Report

A well-written and very comprehensive manuscript.

Perhaps the only thing that could have been different is to also include a separate subsection on the functional aspects of regenerative medicine experiments.  

Cannot find any issues with the quality of the English language. 

Round 2

Reviewer 1 Report

 The authors have sufficiently addressed the raised Issues, which lead to even better quality of manuscript